# Dynamic Model of a Humanoid Exoskeleton of a Lower Limb with Hydraulic Actuators

**DOI:** 10.3390/s21103432

**Published:** 2021-05-14

**Authors:** Sebastian Glowinski, Maciej Obst, Sławomir Majdanik, Barbara Potocka-Banaś

**Affiliations:** 1Department of Mechanical Engineering, Koszalin University of Technology, 75453 Koszalin, Poland; 2Institute of Applied Mechanics, Poznan University of Technology, ul. Jana Pawła II 24, 60965 Poznań, Poland; maciej.obst@put.poznan.pl; 3Department of Clinical and Forensic Toxicology, Pomeranian Medical University in Szczecin, Powstanców Wielkopolskich 72, 70111 Szczecin, Poland; majdanik@pum.edu.pl (S.M.); bpotocka@pum.edu.pl (B.P.-B.)

**Keywords:** exoskeletons, lower limb, mathematical model, simulation, dynamics

## Abstract

Exoskeletons are the mechanical systems whose operation is carried out in close cooperation with the human body. In this paper, the authors describe a mathematical model of the hydraulic exoskeleton of a lower limb. The coordinates of characteristic points of the exoskeleton in the sagittal plane as a function of user height are presented. The mathematical models, kinematics, and kinetics equations were determined. The masses of the actuators and their dimensions were selected based on catalog data. The force distribution in the wearable system during the squat is shown. The proposed models allowed us to determine the trajectory of individual points of the exoskeleton and to determine the forces in hydraulic cylinders that are necessary to perform a specific displacement. The simulation results show that the joint moments depend linearly on actuator forces. The dynamics equations of the wearable system are non-linear. The inertia of the system depends on the junction variables and it proves that there are dynamic couplings between the individual axes of the exoskeleton.

## 1. Introduction

The development of robotics and computing power allows the design of devices for amplifying human power. This subject area is a present-day problem when we observe the processes of socio-cultural changes and the issues of society aging. In the 1950s, only 4.9% of the population lived more than 65 years and nowadays this percentage is about 20%, it is anticipated to be more than 35 % in the year 2050. During human aging, we become less able in terms of the body. Elderly people still want to be active, walk in the mountains, run, etc. The advantages of using modern devices can be seen in many areas of human performance. Exoskeletons are applied in medicine and rehabilitation by physiotherapists and patients. Some replicable movements with high precision can be made by machines which are applied in surgery or operating rooms. Mobile anthropomorphic robots are examples of the abovementioned machines which assist in the operation of human muscles and are called exoskeletons [1,2].

The first designs of devices for assisting human muscles were done in the 1950s at Argonne National Laboratory (USA) [3,4]. In 1965, the General Electric company started investigations on the exoskeleton Hardiman I (Human Augmentation Research and Development Investigation Man) [5,6]. The mass of this device was equal to 680 kg and one upper limb could raise a load of 340 kg. At the end of the 1980s, Jeffrey Moore from Los Alamos Laboratory described a vision of a combination of an exoskeleton with a spacesuit [7].

During the last 20 years, exoskeletons have been developed considerably [8]. This could be done due to the grant aid from military sources like the Defense Advanced Research Projects Agency (*DARPA*). One of the research projects was done by Sarcos Research Corporation from Salt Lake City—the goal of this project was the design of a device which could increase the muscle power of a soldier [9]. This device was called the Wearable Energetically Autonomous Robot (WEAR) and it had an autonomous power supply and hydraulic drive systems. The other project of an exoskeleton for lower limbs, the Human Universal Load Carrier (HULC), was carried out by eksoBionics (Richmond, CA, USA) in cooperation with Lockheed Martin [10,11].

Exoskeletons are also applied in rehabilitation [12,13,14]. Cyberdyne (Tsukuba, Japan) made a series of exoskeletons called the Hybrid Assistive Leg (HAL). The aim of this system is assistance during the movement of elderly people and patients with dysfunction of the lower limbs [15]. In the years 2003–2006, specialists from the Balgrist University Hospital (Zurych, Switzerland) made the manipulator *ARMin* (ARM therapy exoskeleton) [16,17]. This exoskeleton could work in three modes: passive, active and active with an Activities of Daily Living (ADL) program. The other interesting device of movement re-education for people with paresis of the lower limbs is the mobile exoskeleton *eksoGT* made by eksoBionics (Richmond, VA, USA) [18,19]. It was intended for people with different neurological issues like stroke, injury of the spinal cord, brain injury, or multiple sclerosis. This device allows the patient to use maximal power for moving. The physiotherapist can set the mode type and the power value which can assist the patient during walking. This fact is essential for the selection of the therapy [20]. Cui proposed the design and control of a 7-degrees of freedom cable-driven arm exoskeleton allowing for dexterous hand reorientation, which is required in everyday tasks [21].

Exoskeletons are applied in industry as well. An example of such an application is the device called Fortis, made by Lockheed Martin (USA) [22,23]. Hyundai made an exoskeleton, which was made as a chair, called the Hundai Chairless Exoskeleton (H-CEX), and a vest, called the Hundai Vest Exoskeleton (H-VEX) [24]. This device unloads the knee joints of the user. A passive lower exoskeleton can compensate gravity [25]. A position control system and hybrid position–torque control system of an exoskeleton’s joints were proposed by Herbin [26].

The design of a new exoskeleton is a time-consuming process. Designers use various solutions with structure optimization [27,28]. The design method of medical devices was presented in works [29,30,31]. One of the elements of the design process (before prototyping) is a mathematical model which characterizes the kinematics and dynamics of the device. The application of Matlab software allows for the performance of simulations determining the range of motion, forces, and moments during the device’s operation [32,33]. The obtained results allow for the selection of the actuators and making some corrections in the exoskeleton construction [34]. Zhang et al. [35] and Zhou et al. [36] presented the parallel mechanism and optimization of a singularity-free workspace. The solutions are biologically inspired lower limb exoskeletons.

Most of the publications related to exoskeletons do not contain the exact geometric models, only a diagram of the wearable system. Thus, the aim of this research was to create a mathematical model of the hydraulic exoskeleton. Hip, knee, and ankle gait angles were obtained by using inertial measurement units (IMUs) implemented in the proposed model and were presented in previous work [37]. This paper is focused on the dynamical model of a hydraulic exoskeleton. In the first section of this paper, the basic model of the hydraulic exoskeleton in the sagittal plane is presented. Then, the coordinates of the exoskeleton and human leg characteristic points as a mathematical explanation are established step by step. Diagrams of free limbs of the exoskeleton and lower limbs with forces, moments of inertia, and reactive forces are presented. The proposed mathematical models are built as a function of human height. This method allows for determining the location of characteristic points of the wearable system. The next part of the paper is focused on the static analysis of exoskeletons. The equations of the dynamics of the wearable system are presented and the experimental results are described. Finally, a brief conclusion with some limitations of the study is presented.

## 2. Materials and Methods

### 2.1. Exoskeleton Scheme

The model of a hydraulic exoskeleton with three actuators is shown in Figure 1a. A kinematic description and mathematical model with the movement simulation are presented in [37].

### 2.2. Coordinates

The origin of the coordinate system was placed in the ankle joint to determine the forces for rising of the human body from the squat position to the standing position. A kinematic model is shown in Figure 1b. Transformation matrixes can be written in the following form:(1)$$\begin{array}{c} M_{a}=\left[\begin{array}{cc} \cos\theta_{a} & \sin\theta_{a}\\ -\sin\theta_{a} & \cos\theta_{a} \end{array}\right]\;\;\;\;\;\;M_{k}=\left[\begin{array}{cc} \cos\theta_{k} & -\sin\theta_{k}\\ \sin\theta_{k} & \cos\theta_{k} \end{array}\right]\\ M_{h}=\left[\begin{array}{cc} \cos\theta_{h} & \sin\theta_{h}\\ -\sin\theta_{h} & \cos\theta_{h} \end{array}\right] \end{array}$$
Locations of characteristic points can be expressed by the following dependence: (2)$$\begin{array}{cc} K & =\left[\begin{array}{c} 0\\ 0 \end{array}\right]\\ M & =\left[\begin{array}{c} 0\\ b_{f}^{⋆} \end{array}\right]\\ R & =M_{a}\left[\begin{array}{c} -(l_{l}-a_{2}-a_{l})\\ 0 \end{array}\right]\\ H & =M_{a}\left[\begin{array}{c} -(l_{l}-a_{l})\\ 0 \end{array}\right]\\ G & =M_{a}\left[\begin{array}{c} l_{l}\\ 0 \end{array}\right]\\ I & =M_{a}\left[\begin{array}{c} -(l_{l}-a_{l})\\ -b_{t} \end{array}\right]\\ J & =M_{a}\left[\begin{array}{c} -(l_{l}-a_{2}-a_{l})\\ b_{l}^{⋆} \end{array}\right]\\ J^{†} & =M_{a}\left[\begin{array}{c} \left(1-\frac{S_{3}}{l_{a}}\right)\left(-l_{l}+a_{2}+a_{l}\right)\\ \left(1-\frac{S_{3}}{l_{a}}\right)b_{l}^{⋆} \end{array}\right]\\ F & =M_{a}\left(M_{k}\left[\begin{array}{c} -a_{t}\\ 0 \end{array}\right]+\left[\begin{array}{c} -l_{l}\\ 0 \end{array}\right]\right)\\ P & =M_{a}\left(M_{k}\left[\begin{array}{c} -a_{t}\\ c^{⋆} \end{array}\right]+\left[\begin{array}{c} -l_{l}\\ 0 \end{array}\right]\right)\\ P^{†} & =M_{a}\left(M_{k}\left[\begin{array}{c} -a_{t}-S_{1}\\ c^{⋆} \end{array}\right]+\left[\begin{array}{c} -l_{l}\\ 0 \end{array}\right]\right)\\ D & =M_{a}\left(M_{k}\left[\begin{array}{c} -(l_{t}-b_{t}^{⋆})\\ 0 \end{array}\right]+\left[\begin{array}{c} -l_{l}\\ 0 \end{array}\right]\right)\\ E & =M_{a}\left(M_{k}\left[\begin{array}{c} -(l_{t}-b_{t}^{⋆}\\ -b_{t} \end{array}\right]+\left[\begin{array}{c} -l_{l}\\ 0 \end{array}\right]\right)\\ E^{†} & =M_{a}\left(M_{k}\left[\begin{array}{c} -\left(l_{t}-b_{t}^{⋆}-S_{2}\right)\\ -b_{t} \end{array}\right]+\left[\begin{array}{c} -l_{l}\\ 0 \end{array}\right]\right)\\ C & =M_{a}\left(M_{k}\left[\begin{array}{c} -l_{t}\\ 0 \end{array}\right]+\left[\begin{array}{c} -l_{l}\\ 0 \end{array}\right]\right)\\ B & =M_{a}\left(M_{k}\left(M_{h}\left[\begin{array}{c} -a^{⋆}\\ c^{⋆} \end{array}\right]+\left[\begin{array}{c} -l_{t}\\ 0 \end{array}\right]\right)+\left[\begin{array}{c} l_{l}\\ 0 \end{array}\right]\right)\\ A & =M_{a}\left(M_{k}\left(M_{h}\left[\begin{array}{c} -(a^{⋆}+\left(c+c^{⋆}\right)\tan\theta_{p})\\ -c \end{array}\right]+\left[\begin{array}{c} -l_{t}\\ 0 \end{array}\right]\right)+\left[\begin{array}{c} -l_{l}\\ 0 \end{array}\right]\right)\\ G_{l} & =M_{a}\left[\begin{array}{c} -(l_{l}-l_{\overset{´}{s}\overset{´}{c}})\\ 0 \end{array}\right]\\ G_{t} & =M_{a}\left(M_{k}\left[\begin{array}{c} -t_{\overset{´}{s}\overset{´}{c}}\\ 0 \end{array}\right]+\left[\begin{array}{c} -l_{l}\\ 0 \end{array}\right]\right)\\ G_{b} & =M_{a}\left(M_{k}\left(M_{h}\left[\begin{array}{c} -b_{\overset{´}{s}\overset{´}{c}}\\ 0 \end{array}\right]+\left[\begin{array}{c} -l_{t}\\ 0 \end{array}\right]\right)+\left[\begin{array}{c} -l_{l}\\ 0 \end{array}\right]\right) \end{array}$$
where $SC_{l}$—location of center of gravity for shank, $SC_{t}$—location of center of gravity for thigh, $J^{†}$—location of center of gravity for third actuator, $P^{†}$—location of center of gravity for first actuator. If the following parameters are known: angle of initial position (squat), angle of final position (standing position), velocity and acceleration values for start and end of the movement, and time of movement. Then, we can determine the locations of individual points with the application of the forward kinematics [37]. The sum of individual angles in the hip joint $\theta_{h}$, knee joint $\theta_{k}$, and ankle joint $\theta_{a}$ and the sum of individual velocities ${\dot{\theta }}_{h}$, ${\dot{\theta }}_{k}$, ${\dot{\theta }}_{a}$ and angular accelerations ${\ddot{\theta }}_{h}$, ${\ddot{\theta }}_{k}$, ${\ddot{\theta }}_{a}$ can be defined by
(3)$$\begin{array}{ccc} \theta_{hk}=\theta_{h}+\theta_{k}, & \theta_{ka}=\theta_{k}+\theta_{a}, & \theta_{hka}=\theta_{h}+\theta_{k}+\theta_{a},\\ {\dot{\theta }}_{hk}={\dot{\theta }}_{h}+{\dot{\theta }}_{k}, & {\dot{\theta }}_{ka}={\dot{\theta }}_{k}+{\dot{\theta }}_{a}, & {\dot{\theta }}_{hka}={\dot{\theta }}_{h}+{\dot{\theta }}_{k}+{\dot{\theta }}_{a},\\ {\ddot{\theta }}_{hk}={\ddot{\theta }}_{h}+{\ddot{\theta }}_{k}, & {\ddot{\theta }}_{ka}={\ddot{\theta }}_{k}+{\ddot{\theta }}_{a}, & {\ddot{\theta }}_{hka}={\ddot{\theta }}_{h}+{\ddot{\theta }}_{k}+{\ddot{\theta }}_{a} \end{array}$$

If c and s denote cosine and sine functions then the coordinates of mass centers, velocities, and accelerations of the single elements of an exoskeleton with a lower limb can be described by the equations
(4)$$\begin{array}{c} \left[\begin{array}{c} x_{l}\\ y_{l} \end{array}\right]=\left[\begin{array}{c} -c\theta_{a}\\ s\theta_{a} \end{array}\right]\left(l_{l}-l_{\overset{´}{s}\overset{´}{c}}\right)\\ \left[\begin{array}{c} x_{3}\\ y_{3} \end{array}\right]=\left[\begin{array}{c} -\left(l_{l}-a_{2}-a_{l}\right)c\theta_{a}\\ b_{l}^{⋆}s\theta_{a} \end{array}\right]\left(1-\frac{S_{3}}{l_{a}}\right)\\ \left[\begin{array}{c} x_{t}\\ y_{t} \end{array}\right]=\left[\begin{array}{c} c_{t}s\theta_{a}s\theta_{k}-c\theta_{a}\left(l_{l}+c_{t}c\theta_{k}\right)\\ -s\theta_{a}\left(l_{l}-c_{t}c\theta_{a}\right)-c_{t}c\theta_{a}s\theta_{k} \end{array}\right]\\ \left[\begin{array}{c} x_{1}\\ y_{1} \end{array}\right]=\left[\begin{array}{c} -l_{l}c\theta_{a}-\left(a_{t}+S_{1}\right)c\left(\theta_{a}+\theta_{k}\right)\\ l_{l}s\theta_{a}-\left(a_{t}+S_{1}+c^{⋆}s\theta_{k}\right)s\left(\theta_{a}+\theta_{k}\right) \end{array}\right]\\ \left[\begin{array}{c} x_{b}\\ y_{b} \end{array}\right]=\left[\begin{array}{c} -l_{l}c\theta_{a}-l_{t}c\left(\theta_{a}+\theta_{k}\right)-b_{\overset{´}{s}\overset{´}{c}}c\left(\theta_{a}+\theta_{k}+\theta_{h}\right)\\ l_{l}s\theta_{a}-l_{t}s\left(\theta_{a}+\theta_{k}\right)+b_{\overset{´}{s}\overset{´}{c}}s\left(\theta_{a}+\theta_{k}+\theta_{h}\right) \end{array}\right]\\ \left[\begin{array}{c} x_{2}\\ y_{2} \end{array}\right]=\left[\begin{array}{c} \frac{1}{l_{k}}\left[b_{t}s\theta_{k}+\left(S_{2}-l_{k}\right)\left(\left(a_{l}-l_{l}\right)c\theta_{a}-\left(b_{t}^{⋆}-l_{t}\right)c\theta_{a}c\theta_{k}-l_{l}\right)+\right.\\ \left.+b_{t}\left(s\theta_{a}-c\theta_{k}\right)+\left(b_{t}^{⋆}-l_{t}\right)s\theta_{a}s\theta_{k}\right]+\left(a_{l}-l_{l}\right)c\theta_{a}+b_{t}s\theta_{a}\\ \frac{1}{l_{k}}\left[\left(S_{2}-l_{k}\right)s\theta_{a}\left(c\theta_{k}\left(b_{t}^{⋆}-l_{t}\right)-l_{l}+b_{t}s\theta_{k}\right)-\left(a_{l}-l_{l}\right)s\theta_{a}+\right.\\ \left.+b_{t}c\theta_{a}+c\theta_{a}\left(s\theta_{k}\left(b_{t}^{⋆}-l_{t}\right)-b_{t}c\theta_{k}\right)\right]+\left(a_{l}-l_{l}\right)s\theta_{a}-b_{t}c\theta_{a} \end{array}\right] \end{array}$$

The movement trajectories of mass centers of the trunk, thigh, shank, and actuators 1, 2, and 3 are shown in Figure 2. Equation (Equation 4) was applied in order to determine the trajectory, where $t_{0}$—initial position (squat), $t_{k}$—standing position. The arrows indicate the movement direction of mass centers in relation to the coordinate system origin. The highest value of displacement on the x-axis can be observed for the trunk and its value is −0.28 m. The highest value of displacement on the y-axis occurs for actuator 3 which drives the ankle joint and its value is 0.2m. The lowest value of displacement on the x-axis can be observed for the mass center of the shank, its value is 0.072m—this fact is logical because the distance from the origin of the coordinate system is small.

After differentiation of the Equation (Equation 4), we get the velocity coordinates for the individual masses
(5)$$\begin{array}{cc} \left[\begin{array}{c} {\dot{x}}_{l}\\ {\dot{y}}_{l} \end{array}\right]= & \left[\begin{array}{c} {\dot{\theta }}_{a}s\theta_{a}\\ {\dot{\theta }}_{a}c\theta_{a} \end{array}\right]\left(l_{l}-l_{\overset{´}{s}\overset{´}{c}}\right)\\ \left[\begin{array}{c} {\dot{x}}_{3}\\ {\dot{y}}_{3} \end{array}\right]= & \left[\begin{array}{c} \left(l_{l}-a_{2}-a_{l}\right){\dot{\theta }}_{a}s\theta_{a}\\ b_{l}^{⋆}{\dot{\theta }}_{a}c\theta_{a} \end{array}\right]\left(1-\frac{S_{3}}{l_{a}}\right)\\ \left[\begin{array}{c} {\dot{x}}_{b}\\ {\dot{y}}_{b} \end{array}\right]= & \left[\begin{array}{c} l_{l}{\dot{\theta }}_{a}s\theta_{a}+l_{t}\left({\dot{\theta }}_{a}+{\dot{\theta }}_{k}\right)s\left(\theta_{a}+\theta_{k}\right)+\\ +b_{\overset{´}{s}\overset{´}{c}}\left({\dot{\theta }}_{a}+\dot{\theta_{k}}+\dot{\theta_{h}}\right)c\left(\theta_{a}+\theta_{k}+\theta_{h}\right)\\ l_{l}{\dot{\theta }}_{a}c\theta_{a}-l_{t}\left({\dot{\theta }}_{a}+{\dot{\theta }}_{k}\right)c\left(\theta_{a}+\theta_{k}\right)+\\ +b_{\overset{´}{s}\overset{´}{c}}\left({\dot{\theta }}_{a}+\dot{\theta_{k}}+\dot{\theta_{h}}\right)s\left(\theta_{a}+\theta_{k}+\theta_{h}\right) \end{array}\right]\\ \left[\begin{array}{c} {\dot{x}}_{1}\\ {\dot{y}}_{1} \end{array}\right]= & \left[\begin{array}{c} -l_{l}{\dot{\theta }}_{a}c\theta_{a}-c^{⋆}\left({\dot{\theta }}_{k}+\frac{1}{2}{\dot{\theta }}_{a}\right)s\left(\theta_{a}+2\theta_{k}\right)+\\ -S_{1}\left({\dot{\theta }}_{a}+{\dot{\theta }}_{k}\right)c\left(\theta_{a}+\theta_{k}\right)+\frac{1}{2}c^{⋆}{\dot{\theta }}_{a}s\theta_{a}+\\ -a_{t}\left({\dot{\theta }}_{a}+{\dot{\theta }}_{k}\right)c\left(\theta_{a}+\theta_{k}\right)\\ -\left(S_{1}+a_{t}+c^{⋆}s\theta_{k}\right)\left({\dot{\theta }}_{a}+{\dot{\theta }}_{k}\right)c\left(\theta_{a}+\theta_{k}\right)+\\ +l_{l}{\dot{\theta }}_{a}c\theta_{a}-c^{⋆}{\dot{\theta }}_{k}c\theta_{k}s\left(\theta_{a}+\theta_{k}\right) \end{array}\right]\\ \left[\begin{array}{c} {\dot{x}}_{t}\\ {\dot{y}}_{t} \end{array}\right]= & \left[\begin{array}{c} {\dot{\theta }}_{a}s\theta_{a}\left(l_{l}+c_{t}c\theta_{k}\right)+c_{t}\left({\dot{\theta }}_{a}c\theta_{a}s\theta_{k}+{\dot{\theta }}_{k}c\theta_{a}s\theta_{k}+{\dot{\theta }}_{k}c\theta_{k}s\theta_{a}\right)\\ {\dot{\theta }}_{a}s\theta_{a}\left(l_{l}+c_{t}c\theta_{k}\right)+c_{t}{\dot{\theta }}_{a}c\theta_{a}s\theta_{k}+c_{t}{\dot{\theta }}_{k}\left(c\theta_{k}s\theta_{a}+c\theta_{a}s\theta_{k}\right) \end{array}\right]\\ \left[\begin{array}{c} {\dot{x}}_{2}\\ {\dot{y}}_{2} \end{array}\right]= & \left[\begin{array}{c} {\dot{\theta }}_{a}\left(b_{t}c\theta_{a}-\left(a_{l}-l_{l}\right)s\theta_{a}\right)+\left[\left(S_{2}-l_{k}\right)({\dot{\theta }}_{k}s\theta_{a}\left[b_{t}s\theta_{k}+\left(b_{t}^{⋆}-l_{t}\right)c\theta_{k}\right]\right.+\\ \left.-{\dot{\theta }}_{k}c\theta_{a}\left[b_{t}c\theta_{k}-\left(b_{t}^{⋆}-l_{t}\right)s\theta_{k}\right]+{\dot{\theta }}_{a}s\theta_{a}\left[\left(b_{t}^{⋆}-l_{t}\right)c\theta_{k}-l_{l}+b_{t}s\theta_{k}\right]\right.+\\ +\left.b_{t}{\dot{\theta }}_{a}c\theta_{a}+{\dot{\theta }}_{a}c\theta_{a}\left[\left(b_{t}^{⋆}-l_{t}\right)s\theta_{k}-b_{t}c\theta_{k}\right]-\left(a_{l}-l_{l}\right){\dot{\theta }}_{a}s\theta_{a})\right]\frac{1}{l_{k}}\\ {\dot{\theta }}_{a}\left(b_{t}s\theta_{a}+\left(a_{l}-l_{l}\right)c\theta_{a}\right)+\left[\left(S_{2}-l_{k}\right)({\dot{\theta }}_{k}c\theta_{a}\left[b_{t}s\theta_{k}+\left(b_{t}^{⋆}-l_{t}\right)c\theta_{k}\right]\right.+\\ +\left.{\dot{\theta }}_{k}s\theta_{a}\left[b_{t}c\theta_{k}-\left(b_{t}^{⋆}-l_{t}\right)s\theta_{k}\right]+{\dot{\theta }}_{a}c\theta_{a}\left[\left(b_{t}^{⋆}-l_{t}\right)c\theta_{k}-l_{l}+b_{t}s\theta_{k}\right]\right.+\\ -\left.b_{t}{\dot{\theta }}_{a}s\theta_{a}-{\dot{\theta }}_{a}s\theta_{a}\left[\left(b_{t}^{⋆}-l_{t}\right)s\theta_{k}-b_{t}c\theta_{k}\right]-\left(a_{l}-l_{l}\right){\dot{\theta }}_{a}c\theta_{a})\right]\frac{1}{l_{k}} \end{array}\right] \end{array}$$

The velocity values on x- and y-axis for individual point masses are presented in Figure 3. The highest values of velocity (about 1 m/s) were obtained by the actuator for the knee joint on the y-axis. The lowest values of velocity on the x- and y-axis were observed for the mass center of the shank.

Differentiation of Equation (Equation 5) gives us the following acceleration components:
(6)$$\begin{array}{c} \left[\begin{array}{c} {\ddot{x}}_{l}\\ {\ddot{y}}_{l} \end{array}\right]=\left[\begin{array}{c} \left({\ddot{\theta }}_{a}s\theta_{a}+{\dot{\theta }}_{a}^{2}c\theta_{a}\right)\\ \left({\ddot{\theta }}_{a}c\theta_{a}-{\dot{\theta }}_{a}^{2}s\theta_{a}\right) \end{array}\right]\left(l_{l}-l_{\overset{´}{s}\overset{´}{c}}\right)\\ \left[\begin{array}{c} {\ddot{x}}_{3}\\ {\ddot{y}}_{3} \end{array}\right]=\left[\begin{array}{c} \left(l_{l}-a_{2}-a_{l}\right)\left({\ddot{\theta }}_{a}s\theta_{a}+{\dot{\theta }}_{a}^{2}c\theta_{a}\right)\\ b_{l}^{⋆}\left({\ddot{\theta }}_{a}c\theta_{a}-{\dot{\theta }}_{a}^{2}s\theta_{a}\right) \end{array}\right]\left(1-\frac{S_{3}}{l_{a}}\right)\\ \left[\begin{array}{c} {\ddot{x}}_{b}\\ {\ddot{y}}_{b} \end{array}\right]=\left[\begin{array}{c} l_{l}{\ddot{\theta }}_{a}s\theta_{a}+l_{l}{\dot{\theta }}_{a}^{2}c\theta_{a}+l_{t}\left({\ddot{\theta }}_{a}+{\ddot{\theta }}_{k}\right)s\left(\theta_{a}+\theta_{k}\right)+\\ +l_{t}{\left({\dot{\theta }}_{a}+{\dot{\theta }}_{k}\right)}^{2}c\left(\theta_{a}+\theta_{k}\right)+\\ +b_{\overset{´}{s}\overset{´}{c}}\left({\ddot{\theta }}_{a}+{\ddot{\theta }}_{k}+{\ddot{\theta }}_{h}\right)c\left(\theta_{a}+\theta_{k}+\theta_{h}\right)+\\ -b_{\overset{´}{s}\overset{´}{c}}{\left({\dot{\theta }}_{a}+{\dot{\theta }}_{k}+{\dot{\theta }}_{h}\right)}^{2}s\left(\theta_{a}+\theta_{k}+\theta_{h}\right)\\ l_{l}{\ddot{\theta }}_{a}c\theta_{a}-l_{l}{\dot{\theta }}_{a}^{2}s\theta_{a}-l_{t}\left({\ddot{\theta }}_{a}+{\ddot{\theta }}_{k}\right)c\left(\theta_{a}+\theta_{k}\right)+\\ +l_{t}{\left({\dot{\theta }}_{a}+{\dot{\theta }}_{k}\right)}^{2}s\left(\theta_{a}+\theta_{k}\right)+\\ +b_{\overset{´}{s}\overset{´}{c}}\left({\ddot{\theta }}_{a}+{\ddot{\theta }}_{k}+{\ddot{\theta }}_{h}\right)s\left(\theta_{a}+\theta_{k}+\theta_{h}\right)+\\ +b_{\overset{´}{s}\overset{´}{c}}{\left({\dot{\theta }}_{a}+{\dot{\theta }}_{k}+{\dot{\theta }}_{h}\right)}^{2}c\left(\theta_{a}+\theta_{k}+\theta_{h}\right) \end{array}\right]\\ \left[\begin{array}{c} {\ddot{x}}_{t}\\ {\ddot{y}}_{t} \end{array}\right]=\left[\begin{array}{c} \left(l_{l}+c_{t}c\theta_{k}\right)\left[{\ddot{\theta }}_{a}s\theta_{a}+{\dot{\theta }}_{a}^{2}c\theta_{a}\right]+c_{t}{\ddot{\theta }}_{a}c\theta_{a}s\theta_{k}+\\ +c_{t}c\theta_{a}s\theta_{k}\left({\dot{\theta }}_{k}^{2}+{\ddot{\theta }}_{k}\right)+c_{t}{\ddot{\theta }}_{k}c\theta_{k}s\theta_{a}+\\ -c_{t}s\theta_{a}s\theta_{k}\left({\dot{\theta }}_{a}^{2}+{\dot{\theta }}_{k}^{2}\right)-2c_{t}{\dot{\theta }}_{a}{\dot{\theta }}_{k}\left(s\theta_{a}s\theta_{k}-c\theta_{a}c\theta_{k}\right)\\ \left(l_{l}+c_{t}c\theta_{k}\right)\left[{\ddot{\theta }}_{a}s\theta_{a}+{\dot{\theta }}_{a}^{2}c\theta_{a}\right]+c_{t}{\ddot{\theta }}_{a}c\theta_{a}s\theta_{k}+\\ +c_{t}\left(c\theta_{a}s\theta_{k}+{\ddot{\theta }}_{k}c\theta_{k}s\theta_{a}+{\dot{\theta }}_{k}^{2}c\theta_{a}c\theta_{k}\right)+\\ -c_{t}s\theta_{a}s\theta_{k}\left({\dot{\theta }}_{a}^{2}+{\dot{\theta }}_{k}^{2}\right)-2c_{t}{\dot{\theta }}_{a}{\dot{\theta }}_{k}\left(s\theta_{a}s\theta_{k}-c\theta_{a}c\theta_{k}\right) \end{array}\right]\\ \left[\begin{array}{c} {\ddot{x}}_{1}\\ {\ddot{y}}_{1} \end{array}\right]=\left[\begin{array}{c} l_{l}\left({\ddot{\theta }}_{a}c\theta_{a}-{\dot{\theta }}_{a}^{2}s\theta_{a}\right)+\left(a_{t}+S_{1}\right)\left({\ddot{\theta }}_{a}+{\ddot{\theta }}_{k}\right)c\left(\theta_{a}+\theta_{k}\right)+\\ +\frac{1}{2}c^{⋆}\left({\ddot{\theta }}_{a}s\theta_{a}+{\dot{\theta }}_{a}^{2}c\theta_{a}\right)-c^{⋆}\left(\frac{1}{2}{\ddot{\theta }}_{a}+{\ddot{\theta }}_{k}\right)s\left(\theta_{a}+2\theta_{k}\right)+\\ -c^{⋆}\left(\frac{1}{2}{\dot{\theta }}_{a}+{\dot{\theta }}_{k}\right)\left({\dot{\theta }}_{a}+2{\dot{\theta }}_{k}\right)c\left(\theta_{a}+2\theta_{k}\right)+\\ +\left(S_{1}{\dot{\theta }}_{a}+\left(a_{t}+S_{1}\right)\left({\dot{\theta }}_{a}+{\dot{\theta }}_{k}\right)\right)\left({\dot{\theta }}_{a}+{\dot{\theta }}_{k}\right)s\left(\theta_{a}+\theta_{k}\right)\\ \left(a_{t}+S_{1}+c^{⋆}s\theta_{a}\right)\left({\ddot{\theta }}_{k}+{\left({\dot{\theta }}_{a}+{\dot{\theta }}_{k}\right)}^{2}s\left(\theta_{a}+\theta_{k}\right)\right)+\\ -{\ddot{\theta }}_{a}c\left(\theta_{a}+\theta_{k}\right)-c^{⋆}\left({\ddot{\theta }}_{k}c\theta_{k}-{\dot{\theta }}_{k}^{2}s\theta_{k}\right)s\left(\theta_{a}+\theta_{k}\right)\\ +l_{l}\left({\ddot{\theta }}_{a}c\theta_{a}-{\dot{\theta }}_{a}^{2}s\theta_{a}\right)-2c^{⋆}{\dot{\theta }}_{k}\left({\dot{\theta }}_{a}+{\dot{\theta }}_{k}\right)c\theta_{k}c\left(\theta_{a}+\theta_{k}\right) \end{array}\right]\\ \left[\begin{array}{c} {\ddot{x}}_{2}\\ {\ddot{y}}_{2} \end{array}\right]=\left[\begin{array}{c} b_{t}\left({\ddot{\theta }}_{a}c\theta_{a}-{\dot{\theta }}_{a}^{2}s\theta_{a}\right)-\left(a_{l}-l_{l}\right)\left({\ddot{\theta }}_{a}s\theta_{a}+{\dot{\theta }}_{a}^{2}c\theta_{a}\right)+\\ +\left[\left(S_{2}-l_{k}\right)(c\theta_{a}\left[b_{t}\left({\dot{\theta }}_{k}^{2}s\theta_{k}-{\ddot{\theta }}_{k}c\theta_{k}\right)+\left(b_{t}^{⋆}-l_{t}\right)\left({\ddot{\theta }}_{k}s\theta_{k}+{\dot{\theta }}_{k}^{2}c\theta_{k}\right)\right]+\right.\\ +s\theta_{a}\left(b_{t}\left({\ddot{\theta }}_{k}s\theta_{k}+{\dot{\theta }}_{k}^{2}c\theta_{k}\right)+\left(b_{t}^{⋆}-l_{t}\right)\left({\ddot{\theta }}_{k}c\theta_{k}-{\dot{\theta }}_{k}^{2}s\theta_{k}\right)\right)+b_{t}{\ddot{\theta }}_{a}c\theta_{a}+\\ +{\ddot{\theta }}_{a}c\theta_{a}\left(\left(b_{t}^{⋆}-l_{t}\right)s\theta_{k}-b_{t}c\theta_{k}\right)-b_{t}{\dot{\theta }}_{a}^{2}s\theta_{a}-{\dot{\theta }}_{a}^{2}s\theta_{a}\left(\left(b_{t}^{⋆}-l_{t}\right)s\theta_{k}-b_{t}c\theta_{k}\right)+\\ -\left(a_{l}-l_{l}\right)\left({\ddot{\theta }}_{a}s\theta_{a}-{\dot{\theta }}_{a}^{2}c\theta_{a}\right)+2{\dot{\theta }}_{a}c\theta_{a}\left(\left(b_{t}^{⋆}-l_{t}\right){\dot{\theta }}_{k}c\theta_{k}+b_{t}{\dot{\theta }}_{k}s\theta_{k}\right)+\\ -2{\dot{\theta }}_{a}s\theta_{a}\left(\left(b_{t}^{⋆}-l_{t}\right){\dot{\theta }}_{k}s\theta_{k}-b_{t}{\dot{\theta }}_{k}c\theta_{k}\right)+{\ddot{\theta }}_{a}s\theta_{a}\left(\left(b_{t}^{⋆}-l_{t}\right)c\theta_{k}-l_{l}+b_{t}s\theta_{k}\right)+\\ \left.+{\dot{\theta }}_{a}^{2}c\theta_{a}\left(\left(b_{t}^{⋆}-l_{t}\right)c\theta_{k}-l_{l}+b_{t}s\theta_{k}\right))\right]\frac{1}{l_{k}}\\ b_{t}\left({\ddot{\theta }}_{a}s\theta_{a}+{\dot{\theta }}_{a}^{2}c\theta_{a}\right)+\left(a_{l}-l_{l}\right)\left({\ddot{\theta }}_{a}c\theta_{a}-{\dot{\theta }}_{a}^{2}s\theta_{a}\right)+\\ +\left[\left(S_{2}-l_{k}\right)(s\theta_{a}\left[b_{t}\left({\dot{\theta }}_{k}^{2}s\theta_{k}-{\ddot{\theta }}_{k}c\theta_{k}\right)+\left(b_{t}^{⋆}-l_{t}\right)\left({\ddot{\theta }}_{k}s\theta_{k}+{\dot{\theta }}_{k}^{2}c\theta_{k}\right)\right]+\right.\\ -c\theta_{a}\left(b_{t}\left({\ddot{\theta }}_{k}s\theta_{k}{\dot{\theta }}_{k}^{2}c\theta_{k}\right)+\left(b_{t}^{⋆}-l_{t}\right)\left({\ddot{\theta }}_{k}c\theta_{k}-{\dot{\theta }}_{k}^{2}s\theta_{k}\right)\right)+\\ +{\dot{\theta }}_{a}^{2}s\theta_{a}\left(c\theta_{k}\left(b_{t}^{⋆}-l_{t}\right)-l_{l}+b_{t}s\theta_{k}\right)+{\ddot{\theta }}_{a}s\theta_{a}\left(\left(b_{t}^{⋆}-l_{t}\right)s\theta_{k}-b_{t}c\theta_{k}\right)+\\ +b_{t}\left({\ddot{\theta }}_{a}s\theta_{a}+{\dot{\theta }}_{a}^{2}c\theta_{a}\right)+{\dot{\theta }}_{a}^{2}c\theta_{a}\left(\left(b_{t}^{⋆}-l_{t}\right)s\theta_{k}-b_{t}c\theta_{k}\right)+\\ +\left(a_{l}-l_{l}\right)\left({\ddot{\theta }}_{a}c\theta_{a}-{\dot{\theta }}_{a}^{2}s\theta_{a}\right)+2{\dot{\theta }}_{a}{\dot{\theta }}_{k}s\theta_{a}\left(\left(b_{t}^{⋆}-l_{t}\right)s\theta_{k}-b_{t}c\theta_{k}\right)+\\ -{\ddot{\theta }}_{a}c\theta_{a}\left(\left(b_{t}^{⋆}-l_{t}\right)c\theta_{k}-l_{l}+b_{t}s\theta_{k}\right)+\left.2{\dot{\theta }}_{a}{\dot{\theta }}_{k}s\theta_{a}\left(\left(b_{t}^{⋆}-l_{t}\right)c\theta_{k}+b_{t}s\theta_{k}\right)\right]\frac{1}{l_{k}} \end{array}\right] \end{array}$$

The values of acceleration components of mass centers for individual elements of the model are presented in Figure 4.

The component values of angles, velocities, and angular accelerations for single joints during a change in position from squatting to standing upright in 1s are shown in Figure 5a. The displacement, velocity, and acceleration of actuators that drive the joints are illustrated in Figure 5b.

Force distribution in hydraulic exoskeleton during knee bending is shown in Figure 1c. The single parts of limbs and the exoskeleton (these are rigid bodies with homogenous mass distribution) are replaced by the finite set of point masses which are aggregated in the pre-selected points [38]. To determine the characteristic angles, we should use the geometrical dependencies (Equation 7) and pay attention to the selection of proper constant distances between the characteristic points of the limb and exoskeleton. According to Jezierski, the angle for this type of drive is $180^{\circ }$ [39]. For angle values of $0^{\circ }$ and $180^{\circ }$, a problem exists because, for these values, we cannot drive the joint. To prevent this situation, we should use a structure called the four-bar linkage mechanism and we should take into account the predicted load of mechanisms and the selection of proper parameters [40].
(7)$$\begin{array}{cc} \gamma_{1} & =\arctan\left(\frac{c^{⋆}}{l_{t}-a_{t}}\right)\\ \gamma_{2} & =\arctan\left(\frac{b_{t}}{l_{t}-b_{t}^{⋆}-b_{l}^{⋆}}\right)\\ \gamma_{3} & =\arctan\left(\frac{b_{l}^{⋆}}{l_{l}-a_{l}-a_{2}}\right)\\ \alpha_{1}\left(\theta_{h}\right) & =\arccos\left({\bar{CB}}^{2}+{\bar{CP}}^{2}-\frac{{\bar{BP}}^{2}\left(\theta_{h}\right)}{2\cdot \bar{CB}\cdot \bar{CP}}\right)\\ \alpha_{2}\left(\theta_{k}\right) & =\arccos\left({\bar{GI}}^{2}+{\bar{EG}}^{2}-\frac{{\bar{EI}}^{2}\left(\theta_{k}\right)}{2\cdot \bar{GI}\cdot \bar{EG}}\right)\\ \alpha_{3}\left(\theta_{a}\right) & =\arccos\left({\bar{JK}}^{2}+{\bar{KM}}^{2}-\frac{{\bar{JM}}^{2}\left(\theta_{a}\right)}{2\cdot \bar{JK}\cdot \bar{KM}}\right) \end{array}$$
where the value of angles $\gamma_{1},\gamma_{2}$ is constant, whereas $\alpha_{1},\alpha_{2}$ and the distance between points $\bar{BP},\bar{EI}$ are functions of angles in hip and knee joints $\theta_{h},\theta_{k}$.

## 3. Results

The diagram of forces, moments of inertia, and reactive forces for free limbs is presented in Figure 6.

Inclination angles of straight lines of actuator operation $\xi_{F1},\xi_{F2},\xi_{F3}$ są $F_{1},F_{2},F_{3}$ can be calculated by the following equation (we should apply here the coordinates from Equation (Equation 2)):(8)$$\begin{array}{c} \xi_{F1}=\arctan\left(\frac{P(2,1)-B(2,1)}{P(1,1)-B(1,1)}\right)\\ \xi_{F2}=\arctan\left(\frac{E(2,1)-I(2,1)}{E(1,1)-I(1,1)}\right)\\ \xi_{F3}=\arctan\left(\frac{J(2,1)-M(2,1)}{J(1,1)-M(1,1)}\right) \end{array}$$

To determine the distances between the individual masses with characteristic points (hip, knee, and ankle joint) in the function of angles $\theta_{h},\;\theta_{k},\;\theta_{a}$, we should use the dependencies (Equation 4) and (Equation 2), and then we get the vectors:(9)$$\begin{array}{cc} r_{\left(1G\right)} & =\left[\begin{array}{c} x_{1}\left(i\right)-G\left(1,i\right)\\ y_{1}\left(i\right)-G\left(2,i\right) \end{array}\right]\\ r_{\left(2G\right)} & =\left[\begin{array}{c} x_{2}\left(i\right)-G\left(1,i\right)\\ y_{2}\left(i\right)-G\left(2,i\right) \end{array}\right]\\ r_{\left(3K\right)} & =\left[\begin{array}{c} x_{3}\left(i\right)-K\left(1,i\right)\\ y_{3}\left(i\right)-K\left(2,i\right) \end{array}\right] \end{array}$$
where the absolute value of a vector with characteristic points G,K can be calculated as a root of the sum of the component squares *x* and *y*:(10)$$\begin{array}{cc} r_{\left(1G\right)} & =\sqrt{r_{1Gx}^{2}+r_{1Gy}^{2}}\\ r_{\left(2G\right)} & =\sqrt{r_{2Gx}^{2}+r_{2Gy}^{2}}\\ r_{\left(3K\right)} & =\sqrt{r_{3Kx}^{2}+r_{3Ky}^{2}} \end{array}$$

The other values like $r_{\left(bC\right)},\;r_{\left(tG\right)}$, and $r_{\left(lK\right)}$ are constant and depend on the user’s height. The components of force $F_{1},F_{2}$, and $F_{3}$ are projected on the individual axes of a pre-selected coordinate system and have the following form:(11)$$\begin{array}{cc} F_{1x}=F_{1}\cos\xi_{F1},\;\;\;\; & F_{1y}=F_{1}\sin\xi_{F1}\\ F_{2x}=F_{2}\cos\xi_{F2},\;\;\;\; & F_{2y}=F_{2}\sin\xi_{F2}\\ F_{3x}=F_{3}\cos\xi_{F3},\;\;\;\; & F_{3y}=F_{3}\sin\xi_{F3} \end{array}$$

Driving moments M${}_{F1}$($F_{1}$,$\alpha_{1}$), M${}_{F2}$($F_{2}$,$\alpha_{2}$), and M${}_{F3}$($F_{3}$,$\alpha_{3}$) can be calculated as
(12)$$\begin{array}{cc} M_{F1}\left(\theta_{h}\right) & =\frac{F_{1}\left(\theta_{h}\right)\cdot \bar{CB}\cdot \bar{CP}\sin\left(\alpha_{1}\right)}{\sqrt{{\bar{CB}}^{2}+{\bar{CP}}^{2}-2\cdot \bar{CB}\cdot \bar{CP}\cos\left(\alpha_{1}\right)}}\\ M_{F2}\left(\theta_{k}\right) & =\frac{F_{2}\left(\theta_{k}\right)\cdot \bar{GI}\cdot \bar{EG}\sin\left(\alpha_{2}\right)}{\sqrt{{\bar{GI}}^{2}+{\bar{EG}}^{2}-2\cdot \bar{GI}\cdot \bar{EG}\cos\left(\alpha_{2}\right)}}\\ M_{F3}\left(\theta_{a}\right) & =\frac{F_{3}\left(\theta_{a}\right)\cdot \bar{JK}\cdot \bar{KM}\sin\left(\alpha_{3}\right)}{\sqrt{{\bar{JK}}^{2}+{\bar{KM}}^{2}-2\cdot \bar{KJ}\cdot \bar{KM}\cos\left(\alpha_{3}\right)}} \end{array}$$
where the forces $F_{1},F_{2}$, and $F_{3}$ from actuators 1, 2, and 3 can be calculated by
(13)$$\begin{array}{cc} F_{1}\left(\theta_{h}\right) & =M_{F1}\left(\theta_{h}\right)\frac{\sqrt{{\bar{CB}}^{2}+{\bar{CP}}^{2}-2\cdot \bar{CB}\cdot \bar{CP}\cos\left(\alpha_{1}\right)}}{\bar{CB}\cdot \bar{CP}\sin\left(\alpha_{1}\right)}\\ F_{2}\left(\theta_{k}\right) & =M_{F2}\left(\theta_{k}\right)\frac{\sqrt{{\bar{GI}}^{2}+{\bar{EG}}^{2}-2\cdot \bar{GI}\cdot \bar{EG}\cos\left(\alpha_{2}\right)}}{\bar{GI}\cdot \bar{EG}\sin\left(\alpha_{2}\right)}\\ F_{3}\left(\theta_{a}\right) & =M_{F3}\left(\theta_{a}\right)\frac{\sqrt{{\bar{JK}}^{2}+{\bar{KM}}^{2}-2\cdot \bar{KJ}\cdot \bar{KM}\cos\left(\alpha_{3}\right)}}{\bar{JK}\cdot \bar{KM}\sin\left(\alpha_{3}\right)} \end{array}$$

The moments M${}_{F1}$($\theta_{h}$), M${}_{F2}$($\theta_{k}$), and M${}_{F3}$($\theta_{a}$) depend in a linear way on forces $F_{1},F_{2}$, and $F_{3}$ which are generated by the actuators, and depend in nonlinear way on angles $\alpha_{1},\alpha_{2}$, and $\alpha_{3}$. The equilibrium equation of the system presented in Figure 6 in a given time can be written in the following way (after some transformations):(14)$$\begin{array}{cc} C_{x} & =G_{b}-F_{1x}\\ C_{y} & =F_{1y}\\ M_{F1} & =M_{Gb}\\ G_{x} & =C_{x}+F_{1x}+F_{2x}+G_{2}+G_{t}+G_{1}\\ G_{y} & =F_{1y}-C_{y}-F_{2y}\\ M_{F2} & =M_{G2}+M_{Gt}+M_{G1}+M_{C}+M_{F1G}\\ K_{x} & =G_{x}+G_{3}+F_{3x}+G_{l}+F_{2x}\\ K_{y} & =-G_{y}-F_{3y}-F_{2y}\\ M_{F3} & =M_{G3}+M_{Gl}+M_{G}+M_{F2} \end{array}$$
where we can use the equation of state to determine the forces $F_{1},F_{2}$, and $F_{3}$, reactions C,G, and K, and moments in characteristic points of the system:(15)$$\begin{array}{cc} M_{Gb} & =G_{b}r_{\left(bCy\right)}\\ F_{1} & =\frac{G_{b}r_{\left(bCy\right)}}{\left(\sqrt{{\left(c^{⋆}\right)}^{2}+{\left(a^{⋆}\right)}^{2}}\right)\sin\left(\gamma_{1}+\alpha_{1}\right)}\\ C_{x} & =G_{b}-F_{1}\cos\xi_{F1}\\ C_{y} & =F_{1}\sin\xi_{F1}\\ M_{F1} & =M_{Gb}\\ G_{x} & =C_{x}+F_{2}\cos\xi_{F2}+G_{2}+G_{t}+G_{1}\\ G_{y} & =F_{1}\sin\xi_{F1}-C_{y}-F_{2}\sin\xi_{F2}\\ M_{F2} & =M_{G2}+M_{Gt}+M_{G1}+M_{C}+M_{F1G}\\ K_{x} & =G_{x}+G_{3}+F_{3}\cos\xi_{F3}+G_{l}\\ K_{y} & =G_{y}-F_{2}\sin\xi_{F2}-F_{3}\sin\xi_{F3}\\ M_{F3} & =M_{G3}+M_{Gl}+M_{G}+M_{F2} \end{array}$$

Mass, dimensions, and location of the center of gravity for actuators have been defined with information from catalogs [41,42,43]. Every actuator was modeled in SolidWorks and this allowed us to calculate the moments of inertia [44]. The dependencies (Equation 15) are valid for any mass and length of a limb part with the individual elements of the exoskeleton. The orientation of the coordinate system is given in Figure 6c. The movement trajectory of the exoskeleton was divided into one hundred elements where the location of characteristic points, components of forces, and moments were defined. We define the equilibrium conditions by using Equation (Equation 15) and next we calculate the forces in every actuator. From the first equation, we can calculate moment M${}_{Gb}$ with point *C*—it results from the displacement of the center of gravity of the upper part of the body with an exoskeleton in relation to the hip joint. Next, we can determine the force $F_{1}$ and the reaction of the hip joint C${}_{x}$ and C${}_{y}$ which are necessary to balance the moment. The next step is the determination of the moment in the knee joint *G* (we need to take into consideration the moment from the hip joint), force $F_{2}$ for the equilibrium of the system, and reactions G${}_{x}$ and G${}_{y}$. The last step is the calculation of the moment in the ankle joint *K*, force $F_{3}$ in the actuator, and reactions K${}_{x}$, K${}_{y}$ and M${}_{x}$, M${}_{y}$. The abovementioned procedure is repeated for every new position. Force values $F_{1},\;F_{2}$, and $F_{3}$ for actuators with components *x* and *y* and reactions in individual joints are presented in Figure 7. The force value $F_{1y}$ for the final position is 9 N. It results from the final position of the inclination angle of the trunk which is $16^{\circ }$, the center of gravity of the trunk is displaced forwards about 2.8 cm with the hip joint.

Component reaction $G_{x}$ in the hip joint for angle range $\theta_{h}=85^{\circ }$−$45^{\circ }$ has a negative value because of the action line of force $F_{1}$. Longitudinal axes of actuators are not parallel to individual limb elements and the rotation of the limb causes changes in the location of mass centers in relation to individual points. Human muscles do not give any forces in the analyzed model. In reality, the forces in flexor muscles and extensors can considerably reduce the participation of actuators in keeping the system equilibrium. We should note that the force in actuator 2 is acting both on point *E* which is connected with the thigh part of the exoskeleton and point *I* which is connected with the shank part.

Moments of inertia of human body parts in relation to the center of gravity were calculated based on the anthropometric data—the mass of the exoskeleton was divided and assigned to every limb (i.e., trunk with the exoskeleton, head, upper limbs). For a man with a mass of 80 kg and a height of 175 cm, the values of inertia moments are the following: $I_{b}$ = 5.6257 ${\mathrm{kgm}}^{2}$, $I_{t}$ = 0.166 ${\mathrm{kgm}}^{2}$, and $I_{l}$ = 0.0787 ${\mathrm{kgm}}^{2}$. Moments of inertia for actuators were calculated for 3D models in SolidWorks [44]. Based on the movement trajectory of individual elements, we calculated the values of velocity and acceleration for every point in the system.

The equations of the translational motion of a hydraulic exoskeleton with a limb in the sagittal plane and the equations of rotational motion in relation to hip, knee, and ankle joints can be described by using dependencies (Equation 6) in the following way:(16)$$\begin{array}{cc} -C_{x}-F_{1x}+m_{b}g & =m_{b}{\ddot{x}}_{b}\\ C_{y}-F_{1y} & =m_{b}{\ddot{y}}_{b}\\ M_{Gb}+M_{F1} & =I_{hC}{\ddot{\theta }}_{h}\\ -G_{x}+C_{x}+F_{1x}-F_{2x}+\left(m_{1}+m_{2}+m_{t}\right)g & =m_{2}{\ddot{x}}_{2}+m_{t}{\ddot{x}}_{t}+m_{1}{\ddot{x}}_{1}\\ F_{1y}-F_{2y}-C_{y}-G_{y} & =m_{1}{\ddot{y}}_{1}+m_{2}{\ddot{y}}_{2}+m_{t}{\ddot{y}}_{t}\\ M_{G2}+M_{Gt}+M_{G1}+M_{C}+M_{F1}+M_{F2} & =I_{kG}{\ddot{\theta }}_{k}\\ G_{x}+F_{2x}-F_{3x}+\left(m_{3}+m_{l}\right)g & =m_{3}{\ddot{x}}_{3}+m_{l}{\ddot{x}}_{l}\\ K_{y}+F_{2y}+F_{3y}+G_{y} & =m_{3}{\ddot{y}}_{3}+m_{l}{\ddot{y}}_{l}\\ -M_{F2}+M_{F3}-M_{Gl}-M_{G3}+M_{G} & =I_{lK}{\ddot{\theta }}_{a} \end{array}$$
where moments of inertia depend on the distance between point masses (Equation (Equation 4)) and the joint. These can be calculated as:$$\begin{array}{cc} I_{hC}= & I_{b}+m_{b}r_{bC}^{2}=5,6257+0,4402,7=6,0659\;{\mathrm{kgm}}^{2}\\ I_{kG}= & I_{hC}+\left[I_{t}+m_{t}r_{tG}^{2}\right]+\left[I_{1}+m_{1}r_{1G}^{2}\right]+\left[I_{2}+m_{2}r_{2G}^{2}\right]+m_{b}l_{t}^{2}\\ I_{lK}= & I_{kG}+\left[I_{l}+m_{l}r_{lK}^{2}\right]+\left[I_{3}+m_{3}r_{3K}^{2}\right]+\left(m_{b}+m_{1}+m_{2}+m_{t}\right)l_{l}^{2} \end{array}$$

Dynamic equations of motion (Equation 16) can be described in a general matrix form as
(17)$$F=A_{N}{\ddot{x}}_{j}+A_{N}{\ddot{y}}_{j}+A_{B}{\ddot{x}}_{k}+A_{B}{\ddot{y}}_{k}+mg+I\ddot{\theta }$$
where $A_{N}$—matrix of drive mass, $A_{B}$—matrix of mass of human body parts, I—matrix of moments of inertia. The individual equations were transformed by using dependencies (Equation 12)–(Equation 15) due to the size of each abovementioned matrix. On the left side of nine equations, we marked forces and moments of forces as unknown values: $$\begin{array}{ccc} F_{1} & = & \frac{I_{hC}{\ddot{\theta }}_{h}-m_{b}gr_{\left(bCy\right)}}{r_{\left(BCy\right)}\cos\left(\xi_{F1}\right)+r_{\left(BCx\right)}\sin\left(\xi_{F1}\right)}\\ C_{x} & = & m_{b}\left({\ddot{x}}_{b}-g\right)-F_{1}\cos\left(\xi_{F1}\right)\\ C_{y} & = & m_{b}{\ddot{y}}_{b}+F_{1}\sin\left(\xi_{F1}\right)\\ F_{2} & = & \frac{C_{y}r_{\left(CGx\right)}-C_{x}r_{\left(CGy\right)}-g\left(m_{2}r_{\left(2Gy\right)}+m_{t}r_{\left(tGy\right)}+m_{1}r_{\left(1Gy\right)}\right)}{r_{\left(2Gx\right)}\sin\left(\xi_{F2}\right)-r_{\left(2Gy\right)}\cos\left(\xi_{F2}\right)}+\\ \; & \; & +\frac{I_{kG}{\ddot{\theta }}_{k}+F_{1}\left(r_{\left(PGx\right)}\sin\left(\xi_{F1}\right)+r_{\left(PGy\right)}\cos\left(\xi_{F1}\right)\right)}{r_{\left(2Gx\right)}\sin\left(\xi_{F2}\right)-r_{\left(2Gy\right)}\cos\left(\xi_{F2}\right)}\\ G_{x} & = & F_{2}\cos\left(\xi_{F2}\right)+F_{1}\cos\left(\xi_{F1}\right)+\left(m_{1}+m_{2}+m_{t}\right)g+\\ \; & \; & -m_{1}{\ddot{x}}_{1}-m_{2}{\ddot{x}}_{2}-m_{t}{\ddot{x}}_{t}-C_{x} \end{array}$$
(18)$$\begin{array}{ccc} G_{y} & = & F_{1}\cos\left(\xi_{F1}\right)-F_{2}\cos\left(\xi_{F2}\right)-Cy-m_{1}{\ddot{y}}_{1}-m_{2}{\ddot{y}}_{2}-m_{t}{\ddot{y}}_{t}\\ F_{3} & = & \frac{-G_{y}l_{l}\cos\left(\theta_{a}\right)-G_{x}l_{l}\sin\left(\theta_{a}\right)+g\left(m_{3}r_{\left(3Ky\right)}+m_{l}r_{\left(lKy\right)}\right)}{r_{\left(JKx\right)}\sin\left(\xi_{F3}\right)-r_{\left(JKy\right)}\cos\left(\xi_{F3}\right)}+\\ \; & \; & +\frac{I_{lK}{\ddot{\theta }}_{a}+F_{2}\left(r_{\left(IKy\right)}\cos\left(\xi_{F2}\right)+r_{\left(IKx\right)}\sin\left(\xi_{F2}\right)\right)}{r_{\left(JKx\right)}\sin\left(\xi_{F3}\right)-r_{\left(JKy\right)}\cos\left(\xi_{F3}\right)}\\ K_{x} & = & G_{x}+\left(m_{3}+m_{l}\right)g-m_{3}{\ddot{x}}_{3}-m_{l}{\ddot{x}}_{l}+F_{2}\cos\left(\xi_{F2}\right)-F_{3}\sin\left(\xi_{F3}\right)\\ K_{y} & = & -F_{2}\sin\left(\xi_{F2}\right)-F_{3}\sin\left(\xi_{F3}\right)+m_{3}{\ddot{y}}_{3}+m_{l}{\ddot{y}}_{l}-G_{y}\\ M_{x} & = & F_{3}\cos\left(\xi_{F3}\right)\\ M_{y} & = & F_{3}\sin\left(\xi_{F3}\right) \end{array}$$
where $r_{BCx,y}$, $r_{PGx,y}$, $r_{CGx,y}$, $r_{JKx,y}$ are components *x* and *y* for the distances between points $\bar{BC}$, $\bar{PG}$, $\bar{CG}$, $\bar{JK}$. The forces in point *M* (fixing of piston rod of cylinder 3) are $M_{x}=F_{3x}$ and $M_{y}=F_{3y}$. The procedure is the same as in the case of the analysis of static equilibrium of the system. When writing the equations in a computer program, we should pay particular attention to sign correctness where the positive moment is assumed for a right-handed screw rotation. In the dynamic model, we have to take into account moments of inertia that are connected with mass distribution in the body. The simulation results are illustrated in Figure 8. We can observe that the values of forces and reactions are the same for dynamic and static models—during the start and stop of movement for characteristic points.

The system inertia and time of displacement from the initial position to the final one have a deciding impact on necessary forces for acceleration and deceleration of the system. In the dynamic model, the force from human muscles is zero. As was mentioned earlier, the dynamic equations of the hydraulic exoskeleton of a lower limb are nonlinear. The system inertia depends on junction variables and it proves the occurrence of dynamic coupling between individual axes of the exoskeleton. The essential role here is played by the velocity coupling, i.e., centrifugal force and Coriolis force. The current configuration of the exoskeleton influences the gravitational force which affects the device dynamics.

## 4. Discussion

In this work, the authors developed a hydraulic model of an exoskeleton lower limb. We built a geometrical model of a hydraulic exoskeleton based on anthropometrical parameters. The paper presents the mathematical models related to the dynamics of the exoskeleton based on the matrix notation. The proposed mathematical models are built as a function of human height. This method allows for determining the location of characteristic points of the wearable system. This is especially important when designing this type of device. The starting and ending position of the wearable device was calculated with the determination of the movement trajectory based on the method presented in [37]. The proposed mathematical model of each part of the exoskeleton takes into account the weight of an element of the wearable device and each part of the user’s body. The moments of inertia of the human body were based on the literature data. Static analysis of exoskeletons was done in every position, from the initial position (squat) to the final position (standing position). Next, the time of movement was determined. The necessary forces and moments which are needed to make the displacement were determined on the basis of the presented dynamics equations and including the moments of inertia for every element. The correctness of the models was verified by comparing the consistency of the force values at the beginning and the end of the motion with the static and dynamic analysis.

Of course, the proposed model has some limitations. One is the fact that it is developed in the sagittal plane. The real model has not been developed yet, therefore, it will be necessary to make a prototype and verify the proposed models. Much work needs to be done to accurately design the detailed exoskeleton model, analyze its performance, and make a real model. In future work, the authors will collect data from the real wearable system which will be built. To optimize the operation of the exoskeleton, it is also necessary to conduct parametric studies to select the best possible values of the design variables. The current results did not consider the change in actuator mass when filling the actuator chamber.

## Figures and Tables

**Figure 1 sensors-21-03432-f001:**
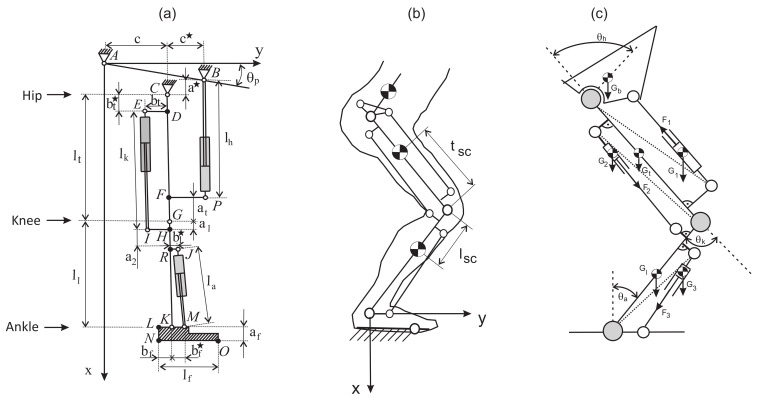
Scheme of the hydraulic exoskeleton in the sagittal plane (**a**), knee bending of the model (**b**), force distribution in hydraulic exoskeleton during knee bending (**c**).

**Figure 2 sensors-21-03432-f002:**
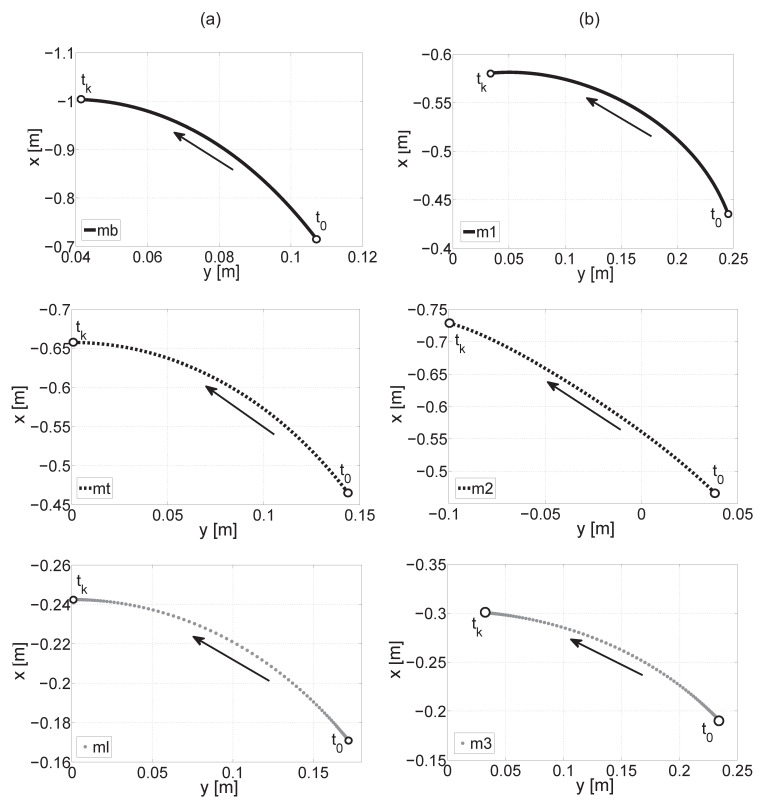
Movement trajectory of the mass center of the trunk, thigh, and shank (**a**), movement trajectory of the mass center of actuators 1, 2, 3 (**b**).

**Figure 3 sensors-21-03432-f003:**
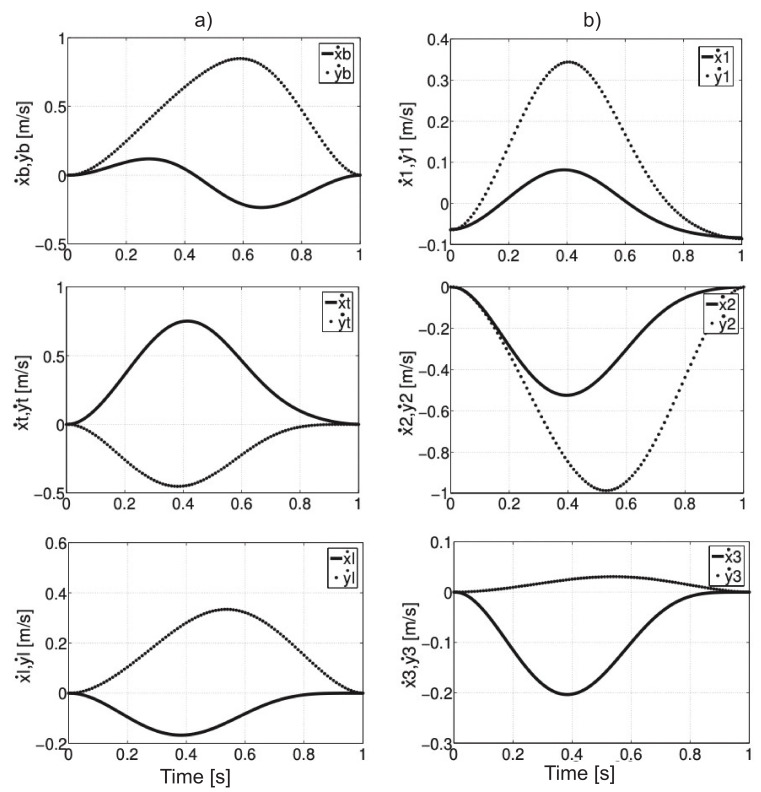
The velocity of the mass center of the trunk, thigh, and shank on *x*- and *y*-axis (**a**), the velocity of the mass center of actuators 1, 2, 3 (**b**).

**Figure 4 sensors-21-03432-f004:**
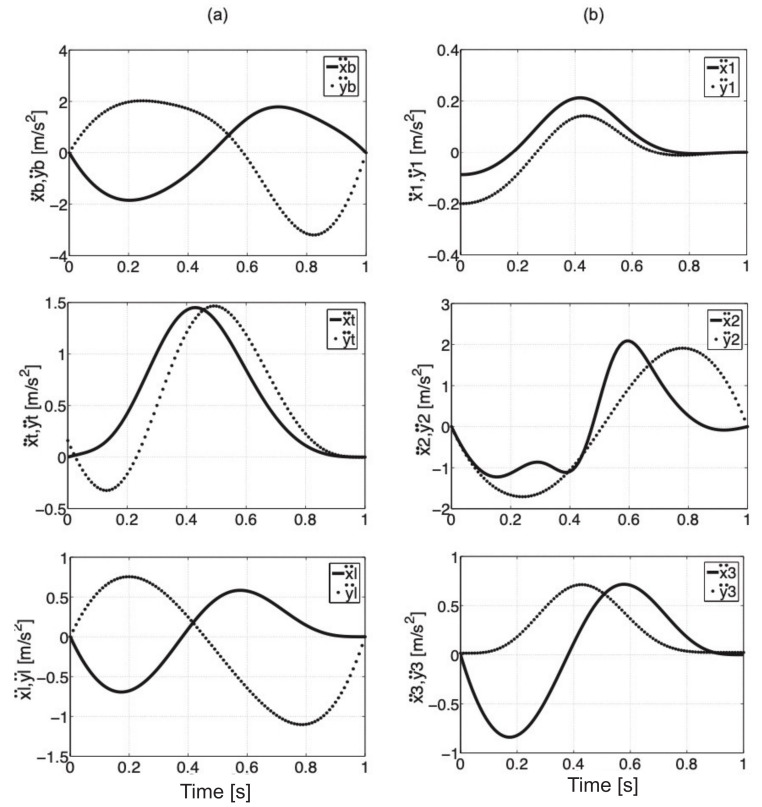
Acceleration values of the mass center of the trunk, thigh, and shank on *x*- and *y*-axis (**a**), acceleration values of the mass center of actuators 1, 2, 3 (**b**).

**Figure 5 sensors-21-03432-f005:**
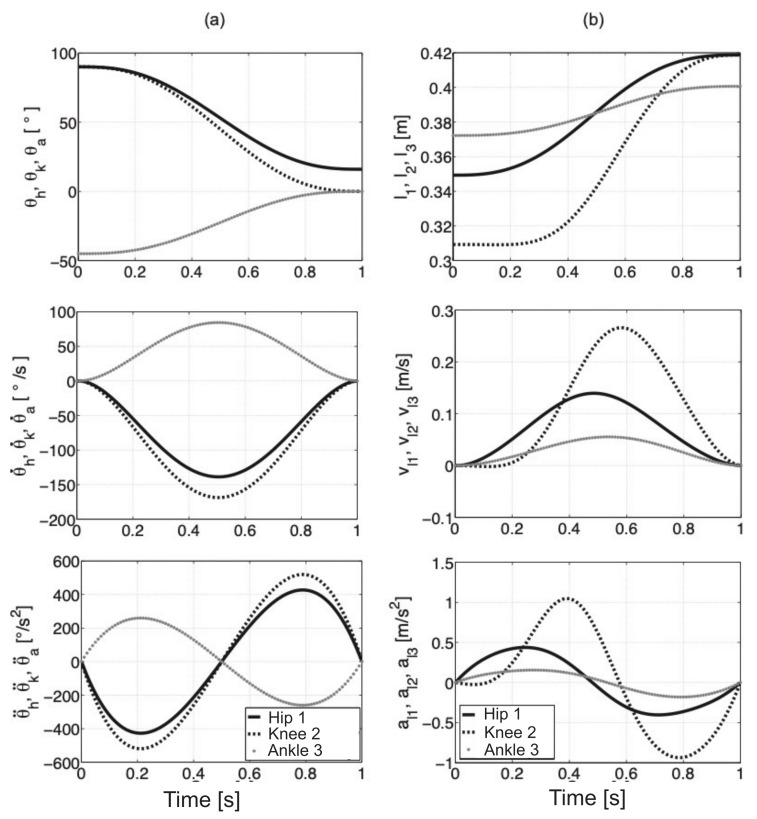
Value variations of angles, velocities, and accelerations for joints (**a**), variations of displacements, velocities, and acceleration for actuators which drive the individual joints (**b**).

**Figure 6 sensors-21-03432-f006:**
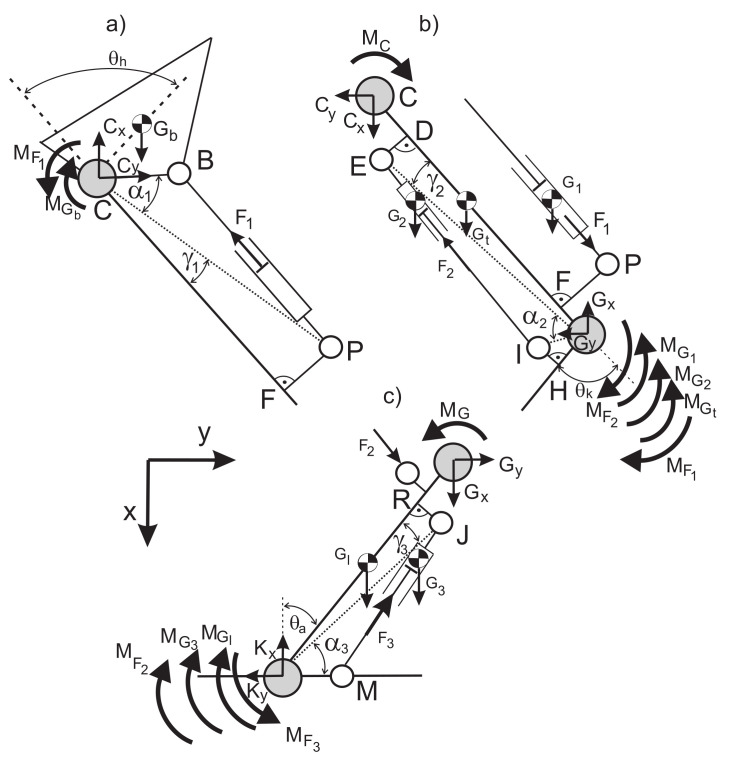
Diagrams of free limbs of the exoskeleton and lower limb with forces, moments of inertia, and reactive forces: (**a**) hip (**b**) knee (**c**) ankle.

**Figure 7 sensors-21-03432-f007:**
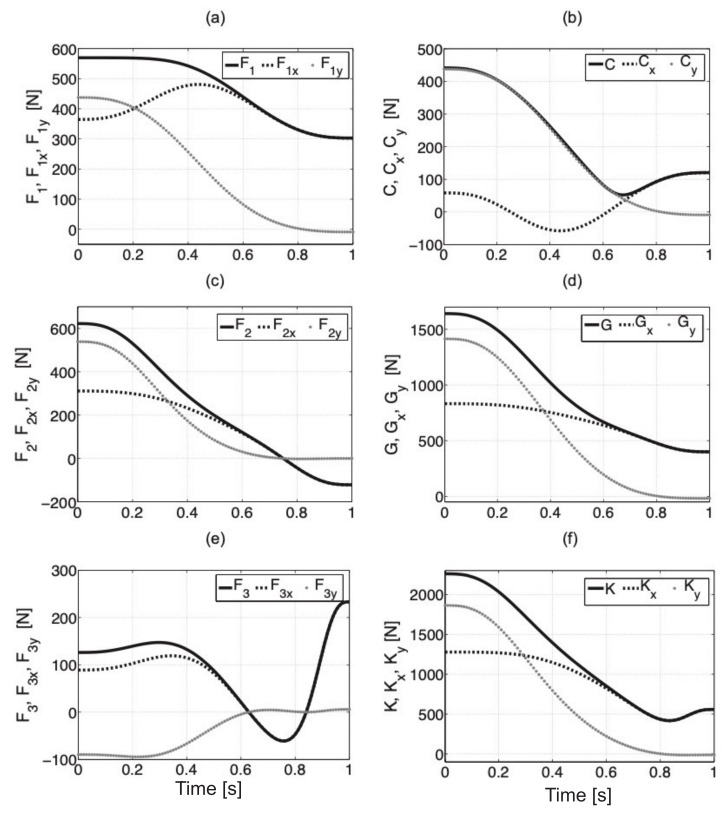
Force values $F_{1}$ (**a**), $F_{2}$ (**c**), and $F_{3}$ (**e**) with components x and y, reactions with components x and y for hip joint (**b**), knee joint (**d**), and ankle joint (**f**).

**Figure 8 sensors-21-03432-f008:**
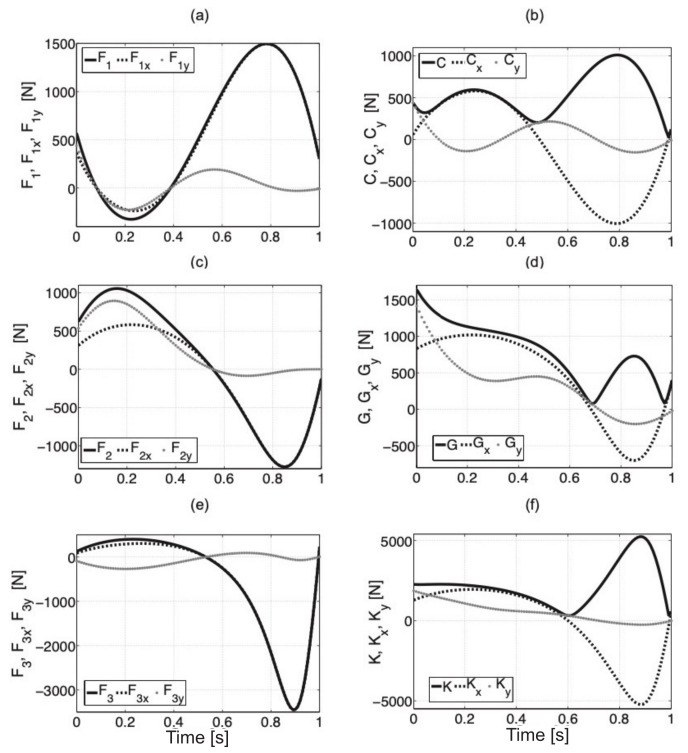
Force values $F_{1}$ (**a**), $F_{2}$ (**c**), $F_{3}$ (**e**) with components x and y in actuators, reactions with components x and y for hip (**b**), knee (**d**), and ankle (**f**) joint.

## Data Availability

Data available on request.

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
