# Peer review of "Dynamic Model of a Humanoid Exoskeleton of a Lower Limb with Hydraulic Actuators"

_sensors, 2021, doi:10.3390/s21103432_

Round 1

Reviewer 1 Report

In this paper a mathematical model of hydraulic exoskeleton of a lower limb is described. However the current work of this manuscript is not comprehensive enough to be accepted for its publication in the journal, and some major improvements is necessary in the content of the different sections.

  1. In section 1, the necessity and innovations of the paper were not described clearly. Only the previous prototypes are described and their disadvantages are not discussed. I think the authors should discuss why they do the research and the differences with the previous dynamic model in details.
  2. In the introduction, some latest references of upper limb and lower limb rehabilitation robots need to be cited. For example:

Design of a 7-DOF cable-driven arm exoskeleton (CAREX-7) and a controller for dexterous motion training or assistance, 2016.

Design of hip joint assistant asymmetric parallel mechanism and optimization of singularity-free workspace, 2018.

A novel precision measuring parallel mechanism for the closed-loop control of a biologically inspired lower limb exoskeleton, 2018.

A review on lower limb rehabilitation exoskeleton robots, 2019.

Optimization of the rotational asymmetric parallel mechanism for hip rehabilitation with force transmission factors. 2020

Design of a passive lower limb exoskeleton for walking assistance with gravity compensation ,2020

Configuration synthesis of variable stiffness mechanisms based on guide-bar mechanisms with length-adjustable links, 2021.

Design and optimisation of load-adaptive actuator with variable stiffness for compact ankle exoskeleton, 2021

Author Response

Dear Reviewer,

The authors would like to thank the anonymous reviewers for their effort in reviewing the manuscript and for their valuable and constructive comments and fruitful observations, which helped in improving the quality of the manuscript to a publishable standard. Detailed below are the responses to the reviewers’ comments and suggestions. Reviewers’ questions and comments are in BLACK and the authors’ answers and comments are in RED (Reviewer 1. docx file).

Reviewer 2 Report

The manuscript lacks a summary of the parameter values of the given equations, which makes it impossible to perform an analysis of the presented solutions. At the stage of conducted research, have you selected the method of control of the modelled exoskeleton?  In the introduction the authors would have mentioned the solutions described in the following exoskeleton publications:

• Anam, Khairul, and Adel Ali Al-Jumaily. “Active Exoskeleton Control Systems: State of the Art.” Procedia Engineering, International Symposium on Robotics and Intelligent Sensors 2012 (IRIS 2012), 41 (January 1, 2012): 988–94. https://doi.org/10.1016/j.proeng.2012.07.273. • Lee, T., Lee, D., Song, B., & Baek, Y. S. (2020). Design and control of a polycentric knee exoskeleton using an electro-hydraulic actuator. Sensors, 20(1), 211. • Herbin, PaweÅ‚, and MirosÅ‚aw Pajor. "The torque control system of exoskeleton ExoArm 7-DOF used in bilateral teleoperation system." AIP Conference Proceedings. Vol. 2029. No. 1. AIP Publishing LLC, 2018. • Shen, Y., J. Ma, B. Dobkin, and J. Rosen. “Asymmetric Dual Arm Approach For Post Stroke Recovery Of Motor Functions Utilizing The EXO-UL8 Exoskeleton System: A Pilot Study.” In 2018 40th Annual International Conference of the IEEE Engineering in Medicine and Biology Society (EMBC), 1701–7, 2018. https://doi.org/10.1109/EMBC.2018.8512665.

These publications describe exoskeleton control systems including. It would be important to take into account the influence of the control method in the dynamic model, especially if you want to use an SAE system. I did not observe in the equations the change in the position of the centre of mass of the hydraulic cylinders and their mass during the change in the piston rod position. For large actuator diameters this can have a significant effect on the dynamic properties of whole exoskeleton. Have you determined the hydraulic pump capacity which is required to generate the presented movements? Finally, it is important to mention the source of the graphs of movement trajectory and velocity presented in the paper.

Author Response

Dear Reviewer,

The authors would like to thank the anonymous reviewers for their effort in reviewing the manuscript and for their valuable and constructive comments and fruitful observations, which helped in improving the quality of the manuscript to a publishable standard. Detailed below are the responses to the reviewers’ comments and suggestions. Reviewers’ questions and comments are in BLACK and the authors’ answers and comments are in RED. Please see attached file (Reviewer 2.docx)

Reviewer 3 Report

The text is written in a relatively clear and legible manner. Graph 1 should be larger and more visible. The literature covers the area required for comprehension of the publication and includes both basic and recent contributions. The presented results are intriguing, and their format is appropriate. Minor grammatical and formal errors can be found in the text.

Overall, I only have a few minor suggestions:

- The publication's structure should be described in the Introduction section.

- At (or before) the first use, each variable or label should be explained.

- The section on Conclusion is missing. I recommend to the authors differentiate the content of the discussion and conclusion sections.

- There is also a lack of future research and gaps in the text.

Author Response

Dear Reviewer,

The authors would like to thank the anonymous reviewers for their effort in reviewing the manuscript and for their valuable and constructive comments and fruitful observations, which helped in improving the quality of the manuscript to a publishable standard. Detailed below are the responses to the reviewers’ comments and suggestions. Reviewers’ questions and comments are in BLACK and the authors’ answers and comments are in RED. Please see attached file (Reviewer 3.docx)

Round 2

Reviewer 1 Report

The authors have attempted to address the comments from this reviewer.

In line 194 -195: Most of the publications related to exoskeletons do not contain the exact geometric models - only a diagram of the wearable system.

I think this is the disadvantages of the previous and it is better to be described in section 1.

And further analysis of this problem may be necessary. I mean why most of the publications related to exoskeletons do not contain the exact geometric models may be analyzed.

Author Response

Dear Reviewer, thank you very much again for this valuable comment. Please see attached response in file Reviewer1.pdf.
